# CNN-Bi-LSTM: A Complex Environment-Oriented Cattle Behavior Classification Network Based on the Fusion of CNN and Bi-LSTM

**DOI:** 10.3390/s23187714

**Published:** 2023-09-06

**Authors:** Guohong Gao, Chengchao Wang, Jianping Wang, Yingying Lv, Qian Li, Yuxin Ma, Xueyan Zhang, Zhiyu Li, Guanglan Chen

**Affiliations:** School of Information Engineering, Henan Institute of Science and Technology, Xinxiang 453003, China; gaoguohong@hist.edu.cn (G.G.); wangchengchao@stu.hist.edu.cn (C.W.); lvyingying@hist.edu.cn (Y.L.); liqian@hist.edu.cn (Q.L.); mayuxin@stu.hist.edu.cn (Y.M.); zhangxueyan@stu.hist.edu.cn (X.Z.); lizhiyu@stu.hist.edu.cn (Z.L.); chenguanglan@stu.hist.edu.cn (G.C.)

**Keywords:** cattle, behavior classification, CNN, Bi-LSTM

## Abstract

Cattle behavior classification technology holds a crucial position within the realm of smart cattle farming. Addressing the requisites of cattle behavior classification in the agricultural sector, this paper presents a novel cattle behavior classification network tailored for intricate environments. This network amalgamates the capabilities of CNN and Bi-LSTM. Initially, a data collection method is devised within an authentic farm setting, followed by the delineation of eight fundamental cattle behaviors. The foundational step involves utilizing VGG16 as the cornerstone of the CNN network, thereby extracting spatial feature vectors from each video data sequence. Subsequently, these features are channeled into a Bi-LSTM classification model, adept at unearthing semantic insights from temporal data in both directions. This process ensures precise recognition and categorization of cattle behaviors. To validate the model’s efficacy, ablation experiments, generalization effect assessments, and comparative analyses under consistent experimental conditions are performed. These investigations, involving module replacements within the classification model and comprehensive analysis of ablation experiments, affirm the model’s effectiveness. The self-constructed dataset about cattle is subjected to evaluation using cross-entropy loss, assessing the model’s generalization efficacy across diverse subjects and viewing perspectives. Classification performance accuracy is quantified through the application of a confusion matrix. Furthermore, a set of comparison experiments is conducted, involving three pertinent deep learning models: MASK-RCNN, CNN-LSTM, and EfficientNet-LSTM. The outcomes of these experiments unequivocally substantiate the superiority of the proposed model. Empirical results underscore the CNN-Bi-LSTM model’s commendable performance metrics: achieving 94.3% accuracy, 94.2% precision, and 93.4% recall while navigating challenges such as varying light conditions, occlusions, and environmental influences. The objective of this study is to employ a fusion of CNN and Bi-LSTM to autonomously extract features from multimodal data, thereby addressing the challenge of classifying cattle behaviors within intricate scenes. By surpassing the constraints imposed by conventional methodologies and the analysis of single-sensor data, this approach seeks to enhance the precision and generalizability of cattle behavior classification. The consequential practical, economic, and societal implications for the agricultural sector are of considerable significance.

## 1. Introduction

The study of cattle behavioral classification holds paramount importance in enhancing animal welfare, optimizing production efficiency, and monitoring health conditions [1,2,3]. Traditional approaches to cattle behavior assessment predominantly involve contact methods. Researchers engage in direct observation of cattle, noting behavioral traits, or employ devices such as ear tag sensors [4]. Employing sensors or cameras facilitates the acquisition of comprehensive behavioral data from cattle across diverse environmental contexts and scenarios. Subsequently, the procured data commonly necessitate preprocessing steps, encompassing tasks such as data cleansing, denoising, and rectification of image or video postures. Furthermore, it becomes imperative to synchronize sensor-generated data with visual information, thereby guaranteeing the steadfastness and integrity of the collected data. Within the realm of cattle behavior classification, the extraction of pertinent features from the amassed data emerges as a pivotal determinant of model efficacy. Sensor-derived data hold the potential to furnish insights into cattle movement patterns, postures, accelerations, and various other intricate particulars.

However, conventional approaches that employ sensors or cameras frequently lean on singular data sources, falling short of capturing the multifaceted information inherent in intricate environments. Moreover, these methodologies can be hindered by the use of manually crafted features during the extraction process. Within complex settings, sensor or camera-centered strategies are susceptible to perturbations arising from variables such as lighting conditions and viewing angles, consequently resulting in a deterioration in classification performance. This methodology is marred by human subjectivity, real-time challenges, and inadequate consideration of animal well-being. Consequently, contactless sensor technology and deep learning methodologies are introduced to enable real-time, precise collection and automated analysis of cattle behavior [5,6,7]. The behavioral attributes of cattle are inherently influenced by environmental conditions and fluctuations in light, leading to noise and data incompleteness, which, in turn, compromise the precision and stability of behavioral classification [8]. The current paradigms for behavioral classification rely on manual feature extraction and design, lacking automation and end-to-end learning [9,10].

To transcend the confines of traditional techniques, the domain of behavior classification has witnessed remarkable advancements through deep learning methodologies. Among these, convolutional neural networks (CNN) and bidirectional long short-term memory network (Bi-LSTM) models have emerged as stalwarts, excelling in processing image and sequential data. CNN proficiently extracts spatial features from images, while Bi-LSTM captures temporal dependencies within sequential data. Motivated by this, we propose a behavioral classification network tailored for complex environments, underpinned by a fusion of CNN and Bi-LSTM. This endeavor seeks to harness the strengths of both models, with the ultimate goal of augmenting the precision and robustness of behavioral classification.

The objective of this paper is to undertake an initial exploration into behavioral classification methods for cattle within intricate scenarios. This will be achieved through the utilization of a deep learning approach that synergizes CNN and Bi-LSTM. The ultimate goal is to attain precise categorization of cattle’s feeding patterns and overall health conditions. Through the delineation and recognition of eight pivotal behaviors, our intention is to elevate the well-being of cattle, optimize breeding productivity, and establish a proficient surveillance mechanism for potential health-related issues. Bi-LSTM is well suited for modeling sequential data, effectively capturing the underlying temporal patterns that drive behaviors for enhanced recognition and classification. The integration of the attention mechanism addresses multiple features that may exist within complex scenes, enabling the model to prioritize relevant features corresponding to the ongoing behavior. This leads to augmented classification accuracy. The incorporation of Bi-LSTM and the attention mechanism accomplishes dual objectives. Firstly, it heightens the model’s sensitivity by capturing the temporal characteristics of cow behavior, thus facilitating a clearer differentiation between diverse behaviors. Secondly, the attention mechanism assigns distinct weights to varying behaviors, thereby enabling more precise discrimination of specific behavioral features. The central contributions of this paper can be summarized as follows:We devised a methodology to gather authentic cattle behavior datasets within intricate environmental settings, effectively defining eight distinct behavioral patterns exhibited by cattle. These defined behaviors serve as the foundation for rigorous classification validation.Our work introduces a cattle behavior classification network by integrating convolutional neural networks (CNN) and bidirectional long short-term memory (Bi-LSTM) architecture. This fusion harnesses spatial attributes and temporal relationships, thereby bolstering the precision of behavior classification.The efficacy of our proposed model is substantiated through comparative experiments, pitting it against conventional methodologies and an array of other deep learning approaches.

The structural framework of this paper is as follows: Section 2 delves into an exploration of related research and the pertinent technical context; Section 3 elucidates our proposed methodology, delineating the cattle behavior classification network; Section 4 is dedicated to showcasing our experimental findings and subsequent analysis; and, in conclusion, Section 5 succinctly summarizes the entirety of the paper while offering a glimpse into potential future research directions. Through this study, we endeavor to provide a dependable and efficient resolution to the challenge of cattle behavior classification, thereby catalyzing the advancement of automation and intelligent practices within the realm of agriculture.

## 2. Materials and Methods

### 2.1. Data Acquisition

The study and experiments were conducted at a ranch located in Yuanyang County. The experimental subjects consisted of 50 mature Simmental cows. The barn was partitioned into a dual area configuration, with a cattle pen serving as the separation. On either side of the pathway, troughs were positioned, while, within the cattle’s resting area, a water trough was situated. The barn itself reached a height of 8 m, and cameras were strategically placed 2.8 m above the ground at each trough’s extremity. These cameras were inclined at a 45° angle to facilitate cattle detection. The bedroom’s entrances on both sides were connected to the cattle yard via cattle fences. This cattle yard was enclosed by further cattle fences. Four cameras were strategically installed in the corners, positioned at a height of 3.5 m from the ground. These cameras were also set at a 45° downward angle to observe the ground. The deployment sites of the cameras are visually represented in Figure 1.

The video data acquired by the camera were delivered to the cloud with a resolution of 640 × 480 pixels. All captured cattle behaviors were used for training and testing, and different levels of occlusion were covered in the data because the actual distribution of the farm was used in the data acquisition process. In our data acquisition scenario, the video acquired through video surveillance undergoes a frame-splitting operation to transform it into frame images. We select both daytime and nighttime as distinct periods, from which we extract four video sequences. Each of these sequences provides a source of image data. To address the comprehensive spectrum of cattle behavior, encompassing eight fundamental behaviors, we meticulously curate 700 image data instances for each behavior. This meticulous approach not only establishes equilibrium within the dataset concerning behavioral categories but also bolsters the model’s generalizability and precision. To enhance the diversity and robustness of our training data, thereby fostering the model’s adaptability to diverse cattle behavior scenarios and elevating its generalization prowess, we employed the subsequent data augmentation methodologies within our dataset:

1. Data shearing: the application of random shearing operations involved cropping specific sections of the images along defined axes, subsequently restoring their proportions through stretching in accordance with the shearing ratio. This strategy catered to the array of angles and postures characteristic of cattle behavior, ultimately furnishing the model with refined recognition capabilities, proficiently adapting to varying posture dynamics.

2. Data flipping: random horizontal or vertical flips were intermittently applied to the images, simulating mirrored scenarios. This augmentation technique effectively addressed cattle behavior detection in multiple orientations, thus enhancing the model’s adeptness in accommodating symmetry and directional nuances.

Data rotation: by introducing randomized rotation operations, the images were subjected to angular adjustments, effectively mimicking varying viewpoints from which cattle behavior might be observed. This augmentation approach significantly contributed to the model’s capacity to proficiently navigate through diverse orientations of cattle behavior, thereby fortifying its ability to remain rotationally invariant.

Given the intricacies arising from obstructions posed by cattle fences and the potential overlapping of body dimensions within the video data acquisition process, we employ the OpenCV library. This toolkit facilitates data manipulation, including data segmentation, scaling, and augmentation. In our pursuit of enriched data diversity, we augment the expanded dataset to encompass a total of 1200 instances for each behavior category. This augmentation strategy is essential to ensure that the model can efficaciously distill image feature insights throughout the training regimen, culminating in a substantial elevation of the model’s holistic performance. In our investigation, we partitioned the cattle behavior dataset into distinct training, validation, and test sets, employing an 8:1:1 ratio for this division. Maintaining consistency across the training, validation, and testing subsets holds paramount importance. This uniformity serves the purpose of facilitating a direct and equitable performance evaluation of diverse models, all operating within identical conditions. Such an approach ensures the attainment of fairness and reliability in the comparative assessment of outcomes across these models. The behavior of the cattle is defined in Table 1.

### 2.2. CNN-Bi-LSTM Model

Illustrated in Figure 2 is the comprehensive framework underpinning the behavioral recognition of beef cattle within intricate environments. The initial step involves importing video files procured from the cattle farm into the OpenCV library, where a dedicated function reads these cattle-related video files. A loop is subsequently defined to meticulously process images frame by frame from the video, generating a series of n video frame images. These images are then subjected to processing through the Visual Geometry Group Network (VGG)16, extracting both corresponding feature maps and feature vector sequences. The subsequent classification procedure is facilitated by the Bi-LSTM framework, with the Softmax function employed for the completion of the classification task, yielding probability scores for each behavioral category. This meticulous approach enhances the precision of cattle behavior classification. Ultimately, the image classification task is seamlessly concluded.

#### 2.2.1. CNN-Based Feature Vector Sequence Extraction

The VGG16 model possesses a robust capability for feature extraction. In this study, when n video frame images are input, feature extraction is carried out through 13 convolutional layers and 5 maximum pooling layers. The convolutional layers perform filtering operations on the input image using convolution operations, enabling the extraction of feature information across various scales and abstraction levels. The maximum pooling layer serves to reduce the spatial dimensions of feature mapping, enhancing the salient features to a certain degree. Extracted image features are then directed to a single fully connected layer for classification. Given the substantial dataset in the cattle classification task, we have opted to retain a single fully connected layer for output. This decision serves to decrease the model’s parameter count and computational complexity by omitting two fully connected layers. Although this reduction slightly compromises feature expressiveness and classification performance, it proves advantageous in terms of computational efficiency. The final outcome encompasses a sequence of feature maps and feature vectors. A depiction of the VGG16 model’s processing can be found in Figure 3.

#### 2.2.2. Bi-LSTM-Based Cattle Behavior Classification Task

The long short-term memory network (LSTM) represents a distinct subtype of recurrent neural networks (RNNs) tailored for sequential data processing. Endowed with formidable memory retention and aptitude for modeling long-term dependencies, LSTM’s distinguishing gating mechanism adeptly tackles the challenges posed by gradient vanishing or explosion when addressing cattle behavior classification tasks. LSTM architecture encompasses two integral components: a memory unit and a gating unit. The memory unit’s role is centered on information storage and transfer, whereas the gating unit orchestrates information flow regulation. An illustrative representation of the LSTM model is presented in Figure 4.

The input gate assumes the responsibility of moderating the degree of incorporation of fresh inputs into the memory cell. This gate operates through a sigmoid function to ascertain the significance of each input. Concurrently, a tanh function is employed to generate potential candidate vectors, facilitating the update of values within the memory cell. The function of the forgetting gate pertains to the determination of information that necessitates erasure from the memory cell. By employing a sigmoid function, which considers inputs and the prior moment’s hidden state, this gate yields a value within the 0 to 1 range. The output gate is entrusted with discerning the segments of the memory cell to transmit to the subsequent moment’s hidden state. The computation of the LSTM process adheres to the principles outlined as shown in Equations (1)–(5).
(1)$$i_{t}=\sigma \left(W_{xi}x_{t}+W_{hi}h_{t-1}+W_{ci}c_{t-1}+b_{i}\right),$$
(2)$$f_{t}=\sigma \left(W_{xf}x_{t}+W_{hf}h_{t-1}+W_{cf}c_{t-1}+b_{f}\right),$$
(3)$$c_{t}=f_{t}c_{t-1}+i_{t}\tanh\left(W_{xc}x_{t}+W_{hc}h_{t-1}+b_{c}\right),$$
(4)$$o_{t}=\sigma \left(W_{xo}x_{t}+W_{ho}h_{t-1}+W_{co}c_{t}+b_{o}\right),$$
(5)$$h_{t}=o_{t}\tanh\left(c_{t}\right),$$
where σ denotes the sigmoid function, i, f, c, and o denote the input gate, the forgetting gate, the cell activation vector, and the output gate, respectively, h denotes the hidden vector, b denotes the bias vector, and W denotes the connection weight between two cells.

Recognizing the perpetual movement of cattle, a strategy is employed to surmount the limitation inherent to LSTM individual cells, which primarily access previous information. To address this, the adoption of a Bi-LSTM method is employed for cattle classification. The Bi-LSTM architecture encompasses both forward and backward LSTMs, functioning in parallel to process input feature vector sequences in opposing directions. This bidirectional approach optimally leverages contextual insights, encompassing not just the attributes of the present time step, but also those of preceding and subsequent time steps. This comprehensive perspective aids in a more profound comprehension and adept capture of the dynamic patterns and enduring dependencies intrinsic to cattle behavior. Initiating this process, the feature vector sequences are introduced into the Bi-LSTM network, which is tailor-made for temporal modeling. Ensuring uniformity, sequence lengths are harmonized at the input layer. This standardization guarantees that the sequence inputs possess equivalent timesteps, thus facilitating consistent processing, so that $x_{1},x_{2},\ldots ,x_{n}$ has the same number of timesteps after being used as the input. Our approach involves two key aspects. Firstly, we meticulously regulate information flow and memory updates through the LSTM layer, encompassing bidirectional LSTMs that amalgamate both forward and inverse outputs. Subsequently, the addition of an attention mechanism module within the attention layer elevates our methodology. Within this module, each LSTM output corresponds to an attention unit, effectively introducing a weighted processing mechanism for the Bi-LSTM output. This strategy effectively concentrates the model’s attention on the most pertinent temporal segments, thereby enhancing the accuracy and robustness of the cattle classification task. Finally, the output layer yields the weighted output vector, denoted as ‘y.’ The configuration of the Bi-LSTM architecture is visually presented in Figure 5.

Given that the cattle classification task encompasses multiple categories, the activation function of choice is the softmax. Employing the softmax activation function facilitates the transformation of the output into a probability distribution conducive to multi-category classification. The output vector is subject to normalization via the softmax function, which systematically assigns each element a probability value within the range of 0 to 1. The softmax function’s computation is delineated as shown in Equation (6).
(6)$$soft\max\left(x_{i}\right)=\frac{e^{y_{t}^{i}}}{\sum_{j=1}^{n}e^{y_{t}^{i}}},$$
where $y_{t}$ is the model output and i is the index of the final identification result; the output with the maximum probability value is used as the ID for cattle classification. If it matches the underlying facts, it is considered as a true result. The opposite is recognized as a wrong result. CNN-Bi-LSTM algorithms are shown in Algorithm 1.
**Algorithm 1** CNN-Bi-LSTM algorithm**1:****INPUT:λ**X // Original video**2:****FUNCTION** CNN-Bi-LSTM () {**3:****WHILE** cap.spend () :**4:**Ret.frame = cap.read()**5:****IF** not ret:**6:**break} // Video frame conversion {**7:**features = (preprocessed_cap.reshape(1,*preprocessed_cap.shape))**8:****RETURN** features.flatten() // VGG16 Extracting feature vectors**9:**feature_sequence = [] // Stored as a sequence of feature vectors }**10:**  { $i_{t}=\sigma \left(W_{xi}x_{t}+W_{hi}h_{t-1}+W_{ci}c_{t-1}+b_{i}\right)$ // Input Gate Calculation**11:**  $f_{t}=\sigma \left(W_{xf}x_{t}+W_{hf}h_{t-1}+W_{cf}c_{t-1}+b_{f}\right)$ // Oblivion Gate Calculator**12:**  $c_{t}=f_{t}c_{t-1}+i_{t}\tanh\left(W_{xc}x_{t}+W_{hc}h_{t-1}+b_{c}\right)$**13:**  $o_{t}=\sigma \left(W_{xo}x_{t}+W_{ho}h_{t-1}+W_{co}c_{t}+b_{o}\right)$ // Output Gate Calculation**14:**$soft\max\left(x_{i}\right)=\frac{e^{y_{t}^{i}}}{\sum_{j=1}^{n}e^{y_{t}^{i}}}$ // Output classification probability }**15:** y = xi // The maximum probability is selected as the output image y**16:** **ELSE****17:**  **CONTINUE****18:** **ENDIF****19:**  }**20:****ENDFOR****21:** }**22:**}

## 3. Results

### 3.1. Experimental Settings and Assessment Indicators

#### 3.1.1. Experimental Environment and Parameter Settings

The operating system used for the experiments is LINUX with 16 GB of RAM, NVIDIA GEFORCE RTX3070 graphics card for the GPU, Core i7 for the CPU, and Intel(R)Core (TM)i7-10750HCPU@2.60 GHz 2.59 GHz processor for the network training configuration, and the Pytorch Deep Learning Framework is used to construct the model. The hyperparameter settings for the training phase are shown in Table 2.

To assess the efficacy of CNN-Bi-LSTM in the cattle behavior classification domain, we have selected a control group of networks, namely, mask region-based convolutional neural network (MASK-RCNN), CNN-LSTM, and EfficientNet-LSTM based on their alignment with the objectives of network similarity, generalizability, and sophistication. The rationale behind this selection is rooted in the multifaceted nature of cattle behavior classification.

MASK-RCNN boasts the capacity for both target detection and instance segmentation. In the context of cattle behavior classification, discerning the various parts and activities of cattle is imperative to decipher their behavioral intricacies [11]. Harnessing MASK-RCNN facilitates accurate extraction of cattle regions, culminating in the acquisition of distinct masks for each region, pivotal for precise cattle classification.

CNN-LSTM, with its robust temporal modeling and feature extraction prowess, stands as a notable choice. The behavior of a cow invariably manifests as a continuous time series marked by a sequence of actions and dynamic transitions [12]. The amalgamation of CNN and LSTM facilitates effective time series modeling, where CNN extracts spatial attributes and LSTM captures temporal dependencies, thereby accomplishing refined classification of bovine behaviors.

EfficientNet-LSTM emerges as an apt fusion of EfficientNet’s image feature extraction competence and LSTM’s temporal modeling capabilities [13]. This amalgamation effectively harnesses image features while maintaining a judicious parameter count and computational load. In the classification of cattle behavior, EfficientNet proficiently extracts salient visual traits from images, while LSTM processes feature sequences and undertakes temporal modeling, culminating in the successful completion of the cattle behavior classification endeavor.

The network parameter settings for the four networks are shown in Table 3.

#### 3.1.2. Assessment of Indicators

To gauge the efficacy of CNN-Bi-LSTM in the context of cattle behavior classification, a comprehensive array of model evaluation metrics is established. These encompass precision rate, recall rate, accuracy rate, and the F1 score. Precision rate pertains to the ratio of samples that the model categorizes as positive, which are indeed authentic positive samples. This metric signifies the model’s accuracy in classifying cattle behavior data. Meanwhile, recall is indicative of the model’s capacity to correctly predict positive samples out of the entirety of such samples. This measure delineates how effectively the model identifies relevant data. Accuracy, a foundational metric, quantifies the ratio of accurately predicted samples within a dataset to the total number of samples. This reflects the overarching precision of the model’s classification endeavor. The F1 score harmonizes accuracy and recall in a weighted amalgamation, encapsulating both the model’s precision and comprehensiveness. It emphasizes the model’s aptitude for precise classification while minimizing the risk of overlooking relevant data. The four model assessment indicators are calculated as shown in Equations (7)–(10).
(7)$$precision=\frac{TP}{TP+FP},$$
(8)$$recall=\frac{TP}{TP+FN},$$
(9)$$accuracy=\frac{TP+TN}{FP+FN+TP+TN},$$
(10)$$F1=\frac{2\times Pre\times Rec}{Pre+Rec},$$
where true positive (TP) denotes the number of true cases, true negative (TN) denotes the number of true negative cases, false positive (FP) denotes the number of false positive cases, and false negative (FN) denotes the number of false negative cases.

### 3.2. Ablation Experiment

To investigate the impact of individual model components on classification performance, we conducted ablation experiments using a self-constructed dataset. Employing the CNN-Bi-LSTM model proposed in this paper as the baseline, the experiments were executed in three stages: first, by eliminating the CNN layer and retaining the Bi-LSTM layer for classification—this confirmed the CNN’s role in extracting feature vector sequences; second, by removing the Bi-LSTM layer and keeping the CNN layer—this assessed the Bi-LSTM’s efficacy in modeling temporal sequences and capturing temporal dependencies; and, finally, by employing a traditional behavioral classification method devoid of both the CNN layer and the Bi-LSTM layer—this was conducted to establish the superior performance of the CNN-Bi-LSTM model in the context of cattle behavior classification.

#### 3.2.1. CNN Ablation

The CNN layer is removed in the CNN-Bi-LSTM model, and the image is converted into a sequence of feature vectors using the traditional feature extractor edge histogram (HTG), which is then input to the Bi-LSTM layer for temporal modeling and classification. The experimental results are shown in Table 4.

The edge histogram serves to elucidate image attributes through the extraction of pertinent data such as edge density and direction within the cattle behavior image. This calculation boasts a level of simplicity and efficiency. However, its exclusive concentration on image edges can engender a decline in model precision and accuracy. Additionally, the model’s adaptability suffers owing to variations in cow posture, angles, and other factors inherent in cattle behavior images. In comparison, CNN exhibits enhanced capacity for information capture and adaptability to image fluctuations when it comes to extracting sequences of feature vectors for cattle behavior classification images.

#### 3.2.2. Bi-LSTM Ablation

The Bi-LSTM layer is removed in the CNN-Bi-LSTM model, and a gated recurrent unit (GRU) is used for temporal modeling and classification of CNN extracted feature vector sequences. The experimental results are shown in Table 5.

The GRU presents a simplified structure with fewer parameters compared to the Bi-LSTM, rendering it computationally efficient and facilitating easier training and tuning. Despite these merits, the GRU’s expressive capacity falls short of the Bi-LSTM due to its absence of an explicit memory unit and sole reliance on a single update gate to manage information transfer. Consequently, when confronting diverse behavioral sequence patterns in cattle, the GRU exhibits limitations in capturing nuances, and its classification of multiple cattle behaviors becomes time-consuming. The GRU is also prone to information attenuation and gradient vanishing, affecting model robustness. In contrast, the Bi-LSTM accounts for both forward and backward directions in modeling input sequences, thus offering a more comprehensive approach to sequence modeling. This feature enables it to better capture time-series information, resulting in superior performance for the task of cattle behavior classification.

#### 3.2.3. CNN and Bi-LSTM Ablation

After removing the CNN and Bi-LSTM layers in the CNN-Bi-LSTM model and extracting the sequence of feature vectors of the image using an edge HTG, the classification output is completed by temporal modeling through GRU. The experimental results are shown in Table 6.

It has been experimentally shown that CNN-Bi-LSTM possesses higher detection accuracy and precision than traditional methods.

### 3.3. Evaluation of Generalization Effects

We take the divided training set and test set as the evaluation subject to assess the generalization effect of the CNN-Bi-LSTM model in terms of accuracy and loss rate. The cross-entropy loss function is used to validate the cross subjects and cross views, and the training and test sets of the cross subjects include 15,493 and 7084 samples, respectively. The training and test sets of the cross view include 13,954 and 5986 samples, respectively. As can be seen in Figure 6, model accuracy increases with the number of training sessions.

The loss rates corresponding to the training and test sets in cross-subject and cross-view iterative training are shown in Figure. As can be seen in Figure 7, the model loss rate decreases with the increase in training times.

To verify the classification performance of the CNN-Bi-LSTM model, eight behavioral classification tests of cows were performed using RNN and CNN-Bi-LSTM on 64, 128, and 256 windows, respectively. The classification accuracy is displayed in the confusion matrix as shown in Figure 8.

The graphs (a, c, e) show the classification accuracies of RNN in 64, 128, and 256 windows; (b, d, f) show the classification accuracies of CNN-Bi-LSTM in 64, 128, and 256 windows. The diagonal lines indicate the classification accuracy for each behavior, with darker colors indicating higher accuracy. In the 64 and 128 windows, it can be seen that the RNN makes an error in classifying the Ruminant (Lying) and Ruminant (Standing) behaviors of the cow and is incorrectly classified as the Lying and Standing behaviors due to the fact that the two behaviors are closer to each other in terms of locomotion. In comparison, CNN-Bi-LSTM has been improved in terms of classification effectiveness with superior results. At a window size of 256, the corresponding increase in spatial resolution resulted in a significant improvement in the accuracy of both models.

### 3.4. Comparison Experiment

Four networks, MASK-RCNN, CNN-LSTM, EfficientNet-LSTM, and CNN-Bi-LSTM, are modeled based on the training set in the self-built dataset of this paper. The training set samples are fed into the model, the predicted outputs of the four models are calculated and compared with the real labels, and then the cross-entropy loss function is used to calculate the loss. The experimental results combining the four model assessment indicators are shown in Table 7. The comparison of the accuracy and loss values of the four models is shown in Figure 9.

The results of the daytime classification of the eight cattle behaviors are plotted in Figure 10.

The results of the eight nocturnal behaviors of cattle classification are shown in Figure 11.

## 4. Discussion

### 4.1. Application of Sensor Technology to Cattle Sorting Tasks

Within the realm of monitoring and categorizing cattle behavior, an extensive inquiry was undertaken by Da S. et al. [14], encompassing a thorough analysis of 17 pertinent research papers. Their examination comprehensively embraced a spectrum of factors, including device utilization, sensor typologies employed, behaviors under scrutiny, preprocessing methodologies, feature extraction techniques, and the classifiers implemented. Noteworthy contributions also arose from Robert B. et al. [15], who substantiated the precision of accelerometers in documenting cattle behavior. Additionally, González L. et al. [16] pioneered an unsupervised classification approach concerning electronic data garnered from motion and GPS sensors affixed to collars, with a pronounced emphasis on data sourced from grazing cattle. Nonetheless, approaches hinging on contact-based data acquisition or behavior tracking may inadvertently overlook the crucial dimension of animal welfare considerations. Within our research ambit, the adoption of non-contact video analysis techniques for data acquisition more aptly aligns with the requisites of animal welfare. Concurrently, Bikker J. et al. [17] delved extensively into the precision and accuracy of three-dimensional accelerometers attached to ear tags. Building upon this, Vazquez D. et al. [18] harnessed triaxial accelerometer data emanating from devices affixed to the necks of cattle to discern penned cattle behavior. Their classification accuracy received affirmation through direct visual surveillance. In a differing vein, Arablouei R. et al. [19] centered their focus on a multi-stage pipeline encompassing preprocessing, feature extraction, and classification. This pipeline was harnessed on embedded sensor node systems, facilitating the inference of behaviors such as rumination and lying. Rigorous cross-validation processes and meticulous scrutiny of the statistical and spectral attributes of accelerometer data underscored their model’s efficacy. Leso L. et al. [20] embarked on an exploration of the AFICollar sensor’s efficacy in capturing feeding and rumination behavior data across diverse feed conditions for dairy cows. Their analysis both underscored the system’s proficiency and accentuated limitations that surfaced in complex environmental scenarios. It is within this intricate landscape that our CNN-Bi-LSTM model assumes relevance, poised to contribute substantially toward mitigating the stated limitations.

Abell K. et al. [21] effectively applied the random forest algorithm to the classification of various cow behaviors, encompassing lying down, standing, and walking. This classification algorithm’s predictive efficacy was substantiated through rigorous data collection procedures, involving meticulous video analysis and accelerometer data compilation. The verification process entailed a comprehensive evaluation, inclusive of comparative analyses with alternative classifiers. Building upon this foundational work, Benaissa S. et al. [22] devised a decision tree algorithm rooted in neck accelerometer data from cows. Tailored for real-time prognostication of feeding behavior, this algorithm’s efficiency underwent thorough validation, thereby allowing comparisons against the benchmark random forest algorithm. Nevertheless, our attempt to replicate this methodology unveiled its unsuitability for addressing other distinct behaviors exhibited by cattle. Pioneering a wireless measurement system, Wang J. et al. [23] harnessed 12 leg tags and 6 positional sensors to emulate cattle movement, thus elucidating cattle behavior patterns. Subsequently, Arcidiacono C. et al. [24] detailed a statistical analysis approach predicated on accelerometer data for prompt identification of cow activities, with a specific focus on feeding and standing behaviors. A novel perspective was put forth by Nasirahmadi A. et al. [25], involving the integration of depth sensors, time-of-flight cameras, and 2D cameras for the comprehensive scrutiny of behaviors displayed by both cows and pigs. By redefining and expanding the spectrum of behaviors, including those not previously encompassed such as rumination and defecation in cattle, these methodologies collectively contribute to the broader comprehension and classification of animal behaviors.

### 4.2. Deep Learning Techniques for Cattle Classification Tasks

In relation to deep learning methodologies, we examined the subsequent strategies. Chen C. et al. [26] conducted a comprehensive evaluation that traced the evolution from conventional cattle behavior recognition to computer vision-based recognition techniques. This assessment particularly scrutinized the advancements stemming from both traditional computer vision methodologies and contemporary deep learning approaches. Qiao Y. et al. [27] innovatively fused the Inception-V3 image feature extraction technique with the Bi-LSTM spatiotemporal information capturing method, resulting in a novel approach for individual cattle recognition. Their proposed methodology underwent validation on a dataset encompassing video footage of 50 cows and demonstrated an impressive recognition accuracy of 91% using a 30-frame video-length model. Li Y. et al. [28] introduced a cattle data architecture revolving around the Inertial Measurement Unit (IMU) to effectively classify a range of behaviors such as feeding, standing, rumination, and walking. While the aforementioned methodologies primarily attained cattle classification through image analysis and intelligent recognition, high-accuracy classification of behaviors was constrained, and multi-behavior scenarios necessitated specific environmental conditions. Our CNN-Bi-LSTM model directly addresses this prevailing limitation. Peng Y. et al. [29] devised an RNN network equipped with LSTM models, employing IMU sensors for the acquisition and classification of cattle behavior. Myat N. et al. [30] proposed an enhanced iteration of the Strong-SORT cattle dataset recognition approach, offering continuous detection and tracking of cattle through the use of an RGB camera. Conversely, Peng Y. et al. [31] harnessed an LSTM-RNN model to identify diverse cow behaviors during the three days preceding calving. This comprehensive classification encompassed behaviors such as feeding, rumination, standing, and lying down. Lastly, Hosseininoorbin S. et al. [32] introduced a deep neural network that integrated a joint time–frequency domain data representation to tackle cattle classification challenges. Synthesizing the merits and unresolved complexities from the aforementioned methods, we constructed a cattle behavior classification model meticulously tailored to intricate scenarios.

Jung D. et al. [33] devised an AI-driven system dedicated to cattle sound classification, detection, and recording. Their approach achieved a sound classification accuracy of 94.18%. Fuentes A. et al. [34] developed a deep-learning-based spatiotemporal hierarchical method for cattle behavior recognition. While their system effectively captured cattle behavior around the clock, it encompassed a more limited set of behavior categories. Wu Y. et al. [35] constructed a deep residual Bi-LSTM model tailored to cattle behavior classification, utilizing window sizes of 64, 128, and 256. Xu B. et al. [36] introduced an economical approach for livestock data counting through a combination of classification and semantic segmentation. By optimizing thresholds for quadcopter-captured images and leveraging feature extraction and MASK-RCNN training, they attained precise livestock classification. Despite achieving heightened detection accuracy, these methods were not exempt from substantial limitations. Achour B. et al. [37] employed a four-layer CNN for individual cattle recognition and image analysis. They explored the synergies between CNN and SVM, as well as various CNN combinations, in order to elevate individual recognition accuracy. Qiao Y. et al. [38] devised a deep learning framework to monitor and classify cattle behavior. This pioneering approach seamlessly integrated the C3D network and ConvLSTM, employing video data to classify cattle behavior such as feeding, moving, and standing. Pavlovic D. et al. [39] introduced a robust deep learning framework tailored for monitoring and classifying cattle behavior. Guo Y. et al. [40] proposed a bigru attention-based strategy for fundamental cow behavior classification, including feeding and standing. In parallel, we have successfully validated the proposed model in this paper, demonstrating its adeptness at detecting a wide array of behaviors in complex environments with remarkable accuracy.

### 4.3. Summary and Discussion

In summary, conventional sensor classification approaches typically hinge upon manually designed feature extraction and the preprocessing of sensor data. These methods necessitate the involvement of experts to handpick and engineer features. However, these approaches exhibit two primary shortcomings. Firstly, they tend to possess inherent limitations in terms of scope [41,42,43]. Secondly, they might overlook latent nonlinear and sophisticated features present within the data. Moreover, the feature selection and design undertaken in these traditional methods often lean towards subjectivity, which can potentially result in an inadequate capture of the intricate nuances encapsulated within the data. This, in turn, leads to an inability to seamlessly adapt to the varying data distributions encountered within cattle classification tasks [44,45]. While machine learning and deep learning techniques have demonstrated prowess in classification endeavors, they grapple with certain shared challenges. The deep learning methodologies outlined earlier may grapple with the predicament of data overfitting, especially when applied to the intricate scenarios presented in this study. Addressing this context, the fusion of CNN and the Bi-LSTM method introduced within this paper presents a novel avenue. This method intrinsically learns features from the raw data, consequently mitigating the inherent subjectivity associated with data feature manipulation. Notably, this approach assimilates both image and time series data, ensuring comprehensive coverage of cattle behavioral characteristics.

## 5. Conclusions

Cattle behavior classification technology holds a prominent position within the realm of smart cattle farming. This paper presents an amalgamation of CNN and Bi-LSTM networks, strategically harnessed to proficiently classify cattle behavior within intricate environments. The approach attains an optimal equilibrium between classification accuracy and computational efficiency. Through a comprehensive analysis encompassing ablation experiments, generalization assessments, and comparative evaluations, the findings are outlined as follows:

(1) Ablation experiments: a series of ablation experiments were conducted, involving CNN, Bi-LSTM, and CNN-Bi-LSTM models, each with three layers. This experimental setup, performed within a consistent environment, offers discerning insights into the performance dynamics of individual modules. The metrics observed subsequent to each module’s ablation offer elucidation on their respective contributions.

(2) Generalization assessment: cross-entropy loss was employed as a yardstick to gauge accuracy and loss rates across diverse scenarios and viewpoints. This methodology serves to evaluate the algorithm’s adeptness in generalizing its learned knowledge. The ensuing evaluation places emphasis on the algorithm’s resilience. The accuracy of the confusion matrix solidifies this notion, corroborating the model’s capacity to accurately discern eight distinct motion behaviors displayed by cattle, namely walking, standing, lying down, defecating, eating, drinking, standing rumination, and lying rumination.

(3) Performance comparison: in comparison with alternative models such as mask-rcnn, CNN-LSTM, and Efficientnet-LSTM, the CNN-Bi-LSTM model exhibits superior performance metrics. With a commendable accuracy rate of 94.3% and a recall rate of 93.4%, its efficacy is underscored.

This study serves as a pivotal resource for the field of cattle behavior classification, notably advancing the domains of smart cattle farming and animal welfare. It is important to highlight that our current emphasis centers primarily on the classification of fundamental cattle behaviors. Anticipating the road ahead, our prospective research trajectory will delve into the detection and classification of intricate cattle behaviors, encompassing actions such as jumping, limping, and other nuanced manifestations.

## Figures and Tables

**Figure 1 sensors-23-07714-f001:**
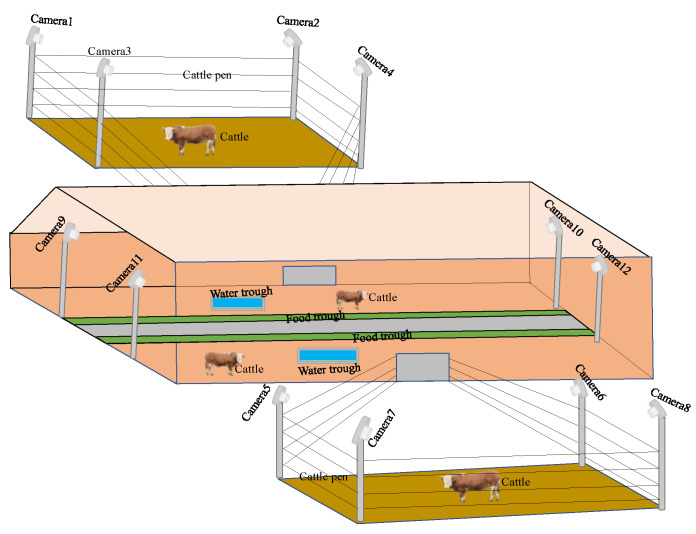
Camera deployment diagram.

**Figure 2 sensors-23-07714-f002:**
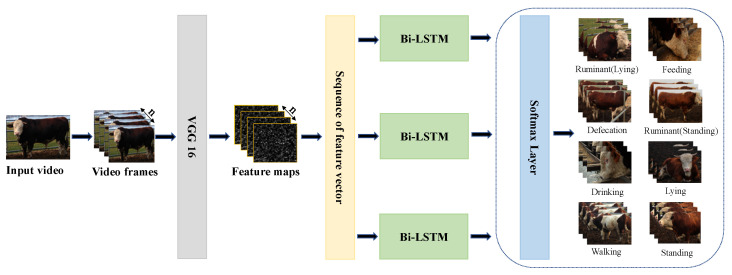
General architecture diagram.

**Figure 3 sensors-23-07714-f003:**
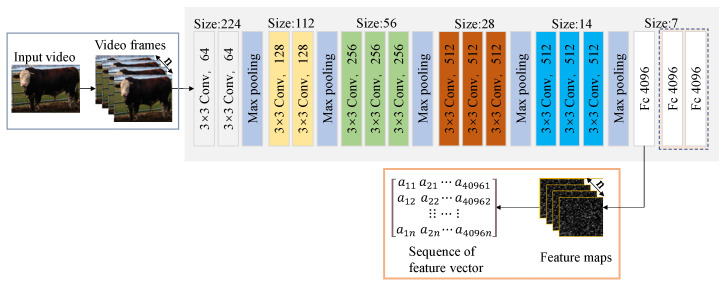
VGG16 flowchart.

**Figure 4 sensors-23-07714-f004:**
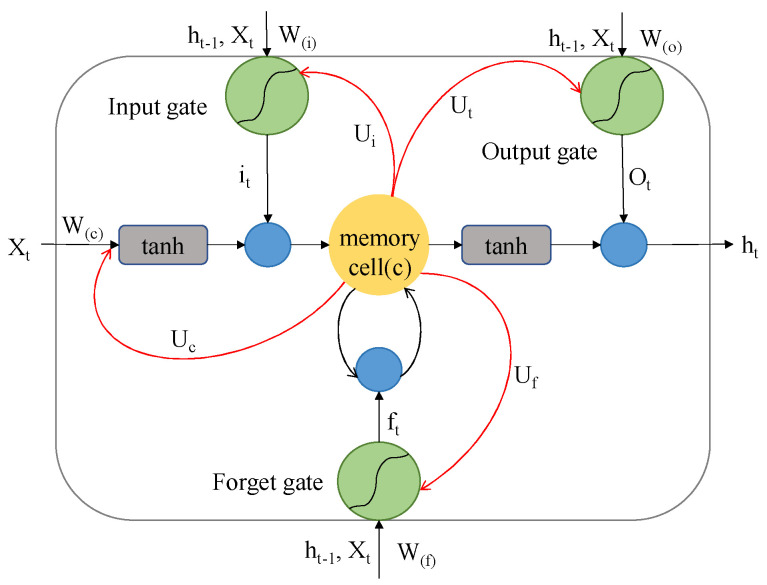
LSTM cell.

**Figure 5 sensors-23-07714-f005:**
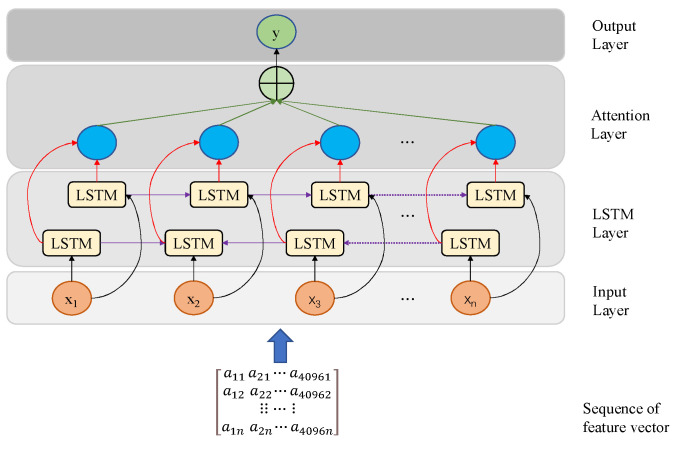
Bi-LSTM network diagram.

**Figure 6 sensors-23-07714-f006:**
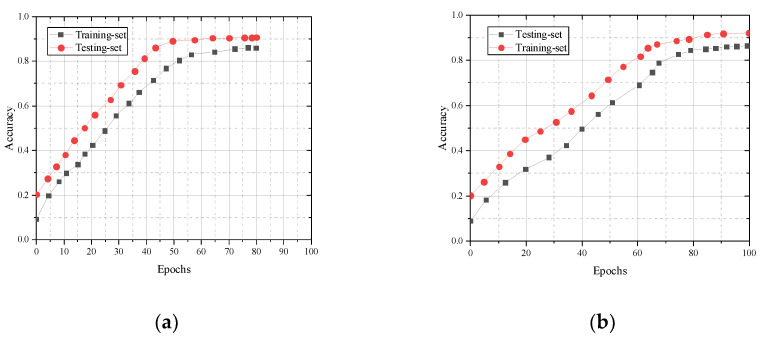
Comparison of training set and test set accuracy: (**a**) cross subject; (**b**) cross view.

**Figure 7 sensors-23-07714-f007:**
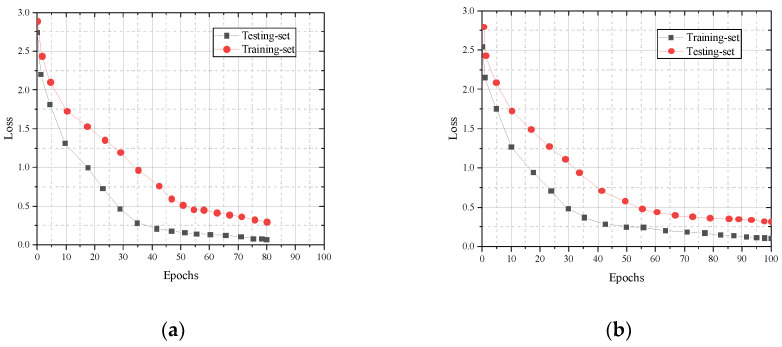
Comparison of LOSS values between the training set and test set: (**a**) cross subject; (**b**) cross view.

**Figure 8 sensors-23-07714-f008:**
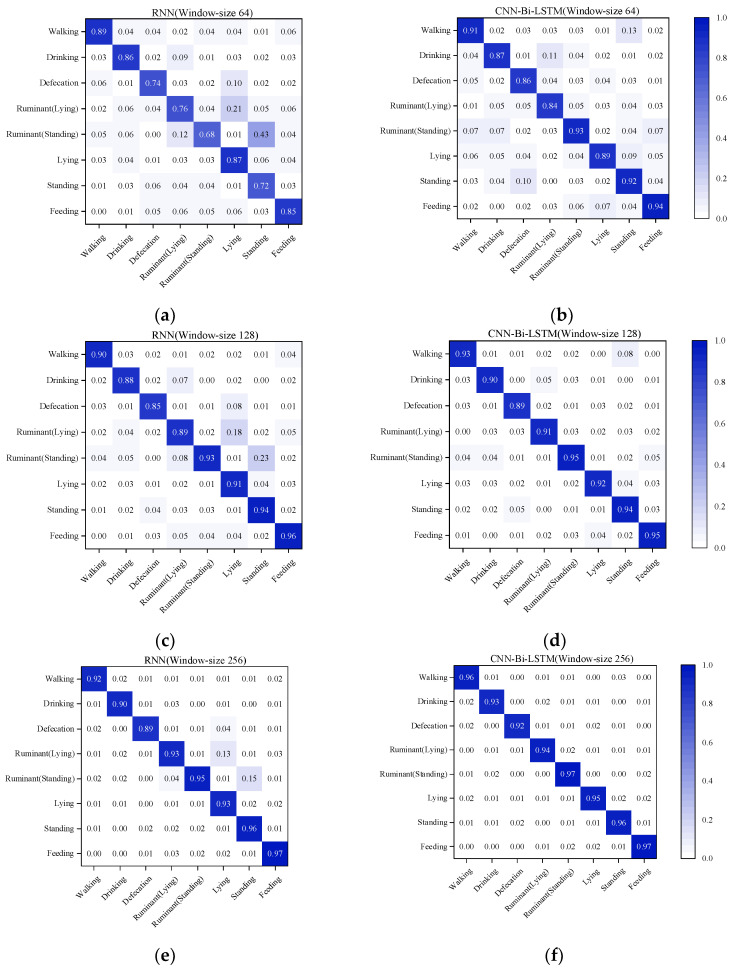
Confusion matrix for classification accuracy of eight cow behaviors. Figure (**a**,**c**,**e**) represent the accuracy of RNN in different windows, and figure (**b**,**d**,**f**) represent the accuracy of CNN-Bi-LSTM in different windows.

**Figure 9 sensors-23-07714-f009:**
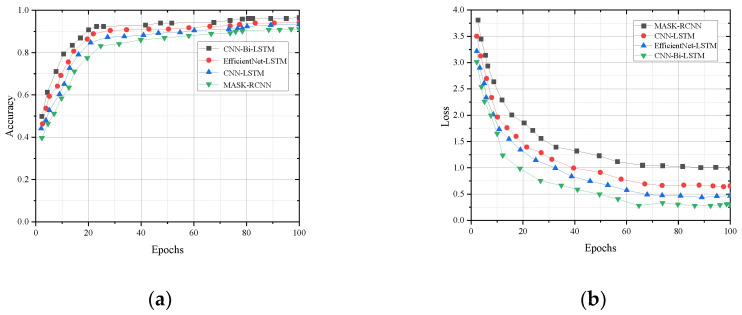
Comparison chart of experimental results: (**a**) accuracy comparison; (**b**) loss comparison.

**Figure 10 sensors-23-07714-f010:**
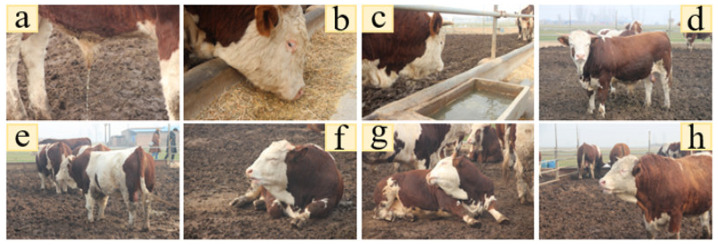
Several basic behaviors: (**a**) defecation, (**b**) feeding, (**c**) drinking, (**d**) standing, (**e**) walking, (**f**) ruminant (lying), (**g**) lying, (h) ruminant (standing).

**Figure 11 sensors-23-07714-f011:**
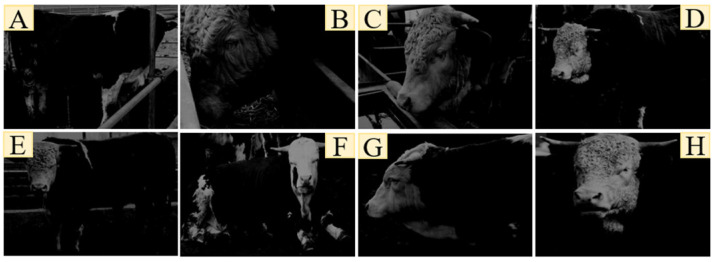
Cattle night behavior classification results: (**A**) defecation, (**B**) feeding, (**C**) drinking, (**D**) walking, (**E**) stand, (**F**) ruminant (lying), (**G**) lying, (**H**) ruminant (standing).

**Table 1 sensors-23-07714-t001:** Definition of cattle behavior.

Behavior	Define
Walking	Cattle standing on all fours and moving their heads and necks.
Drinking	The cattle’s head passes through the cattle pen and comes into contact with the water trough.
Defecation	Excretion by cattle while standing or bending on all fours.
Ruminant (Lying)	Chewing, swallowing, and regurgitation of feed while the cattle are lying on the ground.
Ruminant (Standing)	Chewing, swallowing, and regurgitation of feed while standing on all fours in cattle.
Lying	The cattle’s limbs are bent on the ground.
Standing	Cattle standing upright on all fours in a cattle yard.
Feeding	The head and neck of the cattle pass through the pen and come into contact with the trough.

**Table 2 sensors-23-07714-t002:** Hyper parameter setting.

Parameters	Value
Cuda	10.0.1
Cudnn	10.0.1
Pytorch	1.2.0
Initial Rate	0.01
Termination Rate	0.2
Learning Rate Decay Strategy	Cosine Annealing
Number Of Categories	80
Quantity Per Entry	10
Momentum Parameter	0.9
Input Pixels	299 × 299
Final Decay Rate	4.9 × 10^−4^
Batch	32
Number Of Training	100
Momentum Factor	0.932

**Table 3 sensors-23-07714-t003:** Network parameter setting.

Parameters	Value
Step size	1
Batch size	32
Output Layer Activation Function	Softmax
Loss Function	Cross Entropy Loss
Filling method	Zero-padding
Optimizer	Adam
Learning Rate Scheduler	StepLR scheduler
Hidden Layer Activation Function	Mish
Epoch	100
Filter size	3 × 3

**Table 4 sensors-23-07714-t004:** CNN ablation results.

Model	Precision (%)	Recall (%)	Accuracy (%)	F1 (%)
HTG-Bi-LSTM	90.35	91.02	90.98	91.11
CNN-Bi-LSTM	95.28	96.17	95.62	95.97

**Table 5 sensors-23-07714-t005:** Bi-LSTM ablation results.

Model	Precision (%)	Recall (%)	Accuracy (%)	F1 (%)
CNN-GRU	92.34	92.25	91.25	91.89
CNN-Bi-LSTM	95.31	96.16	95.65	95.92

**Table 6 sensors-23-07714-t006:** CNN and Bi-LSTM ablation results.

Model	Precision (%)	Recall (%)	Accuracy (%)	F1 (%)
HTG-GRU	89.35	90.11	90.03	89.87
CNN-Bi-LSTM	95.29	96.23	95.74	96.05

**Table 7 sensors-23-07714-t007:** Comparison of experimental results.

Model	Evaluation Index (%)			
	Precision	Recall	Accuracy	F1-Score
MASK-RCNN	81.2	78.3	77.5	78.1
CNN-LSTM	85.5	83.9	85.1	84.6
EfficientNet-LSTM	89.8	87.7	90.2	89.6
CNN-Bi-LSTM	94.2	93.4	94.3	94.1

## Data Availability

The data used to support this study are available from the corresponding author upon request.

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
