# Peer review of "CNN-Bi-LSTM: A Complex Environment-Oriented Cattle Behavior Classification Network Based on the Fusion of CNN and Bi-LSTM"

_sensors, 2023, doi:10.3390/s23187714_

Round 1

Reviewer 1 Report

The paper presents a method for Cattle behavior classification based on the fusion of CNN and Bi-LSTM.

The results have been compared with the MASK-RCNN, CNN-LSTM, and EfficientNet-LSTM methods. The method has an accuracy of 94.3%.

The article is well-structured and detailed. There is something unclear in the article.

Why is it important to classify the behavior of cattle? What is the importance of drinking water or being in a state of rest for the agricultural industry?

Is the purpose of the article to provide a preliminary study to determine the feeding behavior or disease states of cattle? It is not clear what the paper aims at. It should be justified to use the eight behaviors defined to improve animal health, increase animal production efficiency and monitor health status.

Finally, the reason why the model is based on the fusion technique should be justified. What is the accuracy level of classical CNN architectures (such as alexnet, mobilenet).

Reviewer 2 Report

This paper proposes and compares a deep learning strategy for cattle behaviour characterization.

The paper seems well written and structured. The literature review seems to be missing some relevant works, as for example: https://www.sciencedirect.com/science/article/pii/S2542660522000415

Also, it is not clear how many pictures were taken, or if the acquisition procedure was design for balancing the classes (Walking, Drinking, etc.). Moreover, I could not found any information regarding eventual data augmentation. Was it needed? If so, what was the procedure to carry out augmentation?

Another aspect that is not clear in the paper (or, better saying, I could not found) regards the design of CNNs benchmark experiments. How was the data splitted? Keeping the same training/validation/testing subsets was a concern to compare the CNNs in the same conditions?

English seems OK. In a later stage of the process, I would recommend to give it final reading for pursuing and correcting typos.

Reviewer 3 Report

-- I suggest letting a person who has been involved in academic papers polish and revise the manuscript before submitting it again due to some errors in grammar and syntax.

-- It would be necessary to refer to the instructions and some published papers in this journal. In particular, the present tense and past tense were misused.

-- The authors failed to cite enough references related to cattle behavior classification using sensors or cameras in the Introduction. Comparing similar research can facilitate readers to know the innovation of the paper.

--Be sure to check the batch, input pixels in table.3, and batch size in table.4. It is not clear the meaning of the number of training. Epoch or iteration?

----The authors didn’t mention how they divide the dataset. Randomly or in other ways? Such information as the number of training and testing data should be in section 2.1. 

----The authors didn’t justify the reason why MASK-RCNN, CNN-LSTM, and EfficientNet-LSTM were selected for comparisons in the manuscript.

-- The author aims to demonstrate what conclusions or findings from Figures 6 and 7. It is evident from the figures that the loss has not converged or stabilized. What does the texting-set mean in the figures?

-- The author has included an extensive list of references in the Discussion section without adding further analysis or discussion. What is the purpose of this?

-- There seems to be a lack of Discussion for the results. It can be improved by further analyzing the observed results and comparing them with similar searches as well as discussing its limitations and future research.

It should be revised and polished before submitted again.

Reviewer 4 Report

This research work presents cattle behavior classification using a fusion of deep learning models. The research area is important and obtained results are also impressive. The following suggestions may be helpful to improve the quality of this paper and must be compliant in order to accept this paper.

1- In the abstract, add a few lines indicating this research's significance of cattle behavior and the technical problem you are addressing. In addition, replace the words smart agriculture with smart cattle farming.

2-There is ambiguity in paragraph section 2.2 (lines 95-98) and long sentences can be split into short sub-sentences 

3- Latest and recent citation of the relevant topic should be added, you can cite the following paper as the method and theme is very similar.

Sheeraz Arif, Jing Wang, Adnan Ahmed Siddiqui.“Bi-directional LSTM with Saliency-aware 3D-CNN features for Human Action Recognition”. Journal of Engineering Research. 9(3A) 1, September 2021, pp. 115-133.

4- The introduction part must be improved cite some relevant and recent research studies and their research gap (if available). In the last part of the introduction define in short why you are introducing Bi-LSTM and attention mechanism. What will be the impact?

5- The last paragraph of the introduction is a chapter or section?

6- Ambiguity in sentence lines 156-157. Re-write the sentence meaningful way

7- Are you sure there is no publically available dataset.? If you are introducing your own dataset so complete details are required in a new sub-section. Explain the size of the dataset, video clips sample related to each behavior, and the dimension of the frames, augmentation methods if you perform to enlarge your dataset. Explain the Split of the samples for training, validation, and testing or you used K-fold?

8-lines 205-221 are not necessary this information is very common you can explain the dataset and their validation process instead

9-Figure 6/7 what is texting?

10-In Fig.8 explain the reason for confusing values in some behavioural categories.

11- The discussion part is too long, you can accomodate the text in sub-sections of experiments.

12- Conclusion is too generic kindly change it to the technical problems and method aspect.

6-

It will be good if this article is reviewed by a native English speaker. There are some grammatical mistakes and ambiguity in some sentences

Round 2

Reviewer 2 Report

The author's reply seems to answer to most of the concerns. Please, just point out, clearly, the augmentations that were used (shear, flips, rotations, or whatever) in your dataset. And, since you suggest the use of OpenCV to perform data augmentation, it would be great to know more details about the process. This is just a technicality, but I think it is important: please, also justify why did you prefer OpenCV over other available libraries that are specialized on doing that kind of operations (e.g. Tensorflow's ImageDataGenerator).    

Reviewer 3 Report

Thanks for your revision and it has been edited better than before. There are still some flaws and limitations that need to be addressed.

--It is suggested to make a summary of cattle behavior classification using sensors or cameras in the Introduction instead of just citing references.

-- Be sure to check the batch size in Table 4, 1024 is too large with an input pixel size of 1024×1024 for the computing resources

--The authors should cite some references when justifying the reason of Mask R-CNN CNN-LSTM, and EfficientNet-LSTM

-- I didn’t get the response to my previous comments “The author aims to demonstrate what conclusions or findings from Figures 6 and 7. It is evident from the figures that the loss has not converged or stabilized. What does the texting-set mean in the figures?”

--The authors should refer to some published papers regarding the tense use for the whole paper.

--The author has once again stacked a series of references in the discussion section. However, this is not the appropriate content for a discussion section. The author should refer to other similar papers to clarify what should be included in this section.

please polish in academic way.

Reviewer 4 Report

The conclusion can be further improved in, future improvement can be added in the conclusion 

English can be further improved
